# 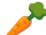 QuaRot: Outlier-Free 4-Bit Inference in Rotated LLMs

**Saleh Ashkboos**
ETH Zurich
saleh.ashkboos@inf.ethz.ch

**Amirkeivan Mohtashami**
EPFL
amirkeivan.mohtashami@epfl.ch

**Maximilian L. Croci**
Microsoft Research
mcroci@microsoft.com

**Bo Li**
ETH Zurich
bolibo@ethz.ch

**Pashmina Cameron**
Microsoft
pcameron@microsoft.com

**Martin Jaggi**
EPFL
martin.jaggi@epfl.ch

**Dan Alistarh**
IST Austria & NeuralMagic
dan.alistarh@ist.ac.at

**Torsten Hoefler**
ETH Zurich
torsten.hoefler@inf.ethz.ch

**James Hensman**
Microsoft Research
jameshensman@microsoft.com

## Abstract

We introduce QuaRot, a new *Qua*ntization scheme based on *Rot*ations, which is able to quantize LLMs end-to-end, including all weights, activations, and KV cache in 4 bits. QuaRot rotates LLMs in a way that removes outliers from the hidden state without changing the output, making quantization easier. This *computational invariance* is applied to the hidden state (residual) of the LLM, as well as to the activations of the feed-forward components, aspects of the attention mechanism, and to the KV cache. The result is a quantized model where all matrix multiplications are performed in 4 bits, without any channels identified for retention in higher precision. Our 4-bit quantized LLAMA2-70B model has losses of at most 0.47 WikiText-2 perplexity and retains 99% of the zero-shot performance. We also show that QuaRot can provide lossless 6 and 8 bit LLAMA-2 models without any calibration data using round-to-nearest quantization. Code is available at github.com/spcl/QuaRot.

## 1 Introduction

Large language models (LLMs) have become increasingly important due to their countless applications. However, using these models in practice, known as inference, requires a significant amount of computation, memory, and energy, specifically during the *prefill* phase, in which the model is supposed to process large prompts and cache them in each layer. Quantization is among the most important techniques to improve both memory and compute issues by keeping the data types at lower precision during the forward pass.

As the prefill stage is known to be compute-bound [Ashkboos et al., 2023], joint quantization aims to reduce the precision of parameters and KV cache (which results in lower memory usage) as well as inputs (known as activations) and compute the forward pass in low precision. However, quantizing the activations is hard as they have large outlier elements (see Figure 1 for an illustrative example) with much larger values, making activation quantization more difficult than weight quantization, especially for the 4-bit case. Previous work relies on using a calibration set to characterize the outlier features and keeping them in higher precision for inference [Zhao et al., 2023, Ashkboos et al., 2023].

38th Conference on Neural Information Processing Systems (NeurIPS 2024).

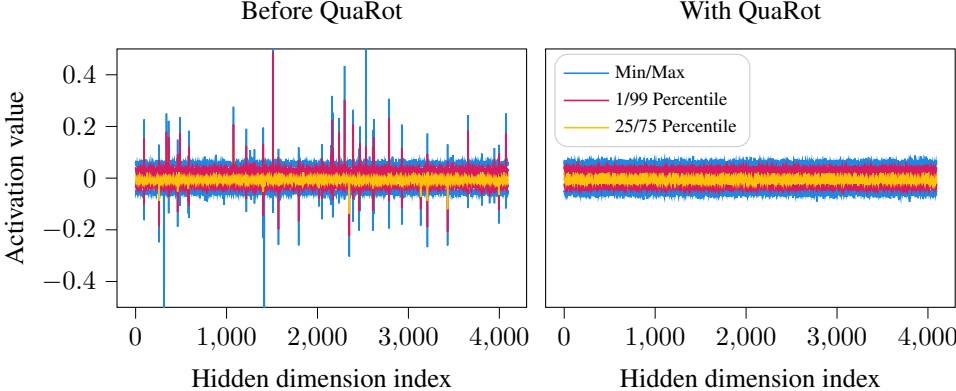

Figure 1: The distributions of activations at the input to the FFN block in LLAMA2-7B model, in the tenth layer. Left: using the default configuration as downloaded from Hugging Face. Right: after processing using QuaRot. The processed distribution has no outliers, leading to superior quantization.

In this work, we address the issue of outlier features by rotating the inputs of the model using randomized Hadamard transformations. We do this using the *computational invariance* idea [Ashkboos et al., 2024] and fuse Hadamard transformations into the weight matrices, resulting in an equivalent network without outlier features. This enables the weights, activations, and KV caches to be quantized to 4 bits with minimal accuracy drop. Our main contributions are:

- We show that randomized Hadamard transformations can be applied to the weight matrices without additional model modifications. In turn, this completely eliminates outlier features and makes the activations easy to quantize, without changing the output of the model. This can be seen as an extension of the *computational invariance* idea, proposed in SliceGPT [Ashkboos et al., 2024] in the context of structured pruning.

- We extend this approach to apply *online* Hadamard transformations to the attention module to remove outlier features in keys and values, enabling the KV cache to be quantized.

- Using the above modifications, QuaRot enables 4-bit LLM inference by quantizing all weights, activations, and KV caches using integer quantization. We provide efficient kernel support for QuaRot: on a LLAMA2-70B model, QuaRot achieves up to $3.33\times$ prefill speedups (on a batch size 64 with 2048 sequence length), and $3.89\times$ memory saving during the decoding stage, with at most 0.47 WikiText-2 perplexity loss. QuaRot preserves 99% of the accuracy of zero-shot tasks and we show that our 6 and 8-bit quantization is lossless with simple round-to-nearest quantization.

## 2   Related Work

The majority of quantization schemes focus on compressing LLMs by using *weight-only quantization*, [Frantar et al., 2022, Dettmers et al., 2023, Lin et al., 2023, Egiazarian et al., 2024, Tseng et al., 2024]. These methods downcast each weight into a low-precision representation and upcast it before the actual computation. The main computation is still performed in high precision. Several works show that, unlike weights, quantizing the activations is hard due to the outlier features [Wei et al., 2022, Dettmers et al., 2022, Xiao et al., 2023]. For 8-bit case, LLM.int8() [Dettmers et al., 2022] identifies the outlier features during inference and keeps them in 16 bits which results in poor performance. SmoothQuant [Xiao et al., 2023] normalizes the features using some scaling factors from a calibration set, solving the issue for the 8-bit case at the cost of introducing extra hyper-parameters. For 4-bit quantization, recent studies identify the outlier features offline and keep them in high precision. Atom [Zhao et al., 2023] developed a complex kernel for mixed-precision MatMul in the presence of outliers while QUIK [Ashkboos et al., 2023] keeps the down-projection layer in 8 bits.

Two weight-only quantization methods, QuIP [Chee et al., 2024] and QuIP# [Tseng et al., 2024] have previously considered improving quantization by applying rotations. Chee et al. [2024] introduced the idea of *incoherence processing* which applies rotation matrices to the left and right of each weight

matrix, as well as the Hessian, which is used in minimizing the weight-quantization objective. Xi et al. [2023] uses a similar idea during training, using exact Hadamard transformations for each linear layer in the forward pass.

Finally, KV cache quantization is another line of research that aims to compress the cached keys and values during the generation phase. This is crucial for large batch size and long-context length generation as the KV cache will be the main memory bottleneck in such problems. Sheng et al. [2023] quantizes the KV cache using 4-bit group-wise quantization. KVQuant [Hooper et al., 2024] pushes this limit to 3-bit quantization and KIVI [Liu et al., 2024] shows promising results on 2-bit KV cache quantization. Such methods show that outliers also exist in the keys, and apply a set of complex ideas (like feature-wise quantization, non-uniform representation, and keeping high precision outliers) to recover the accuracy of a quantized KV cache.

In this work we also adopt the Hadamard transform to improve quantization of weights through incoherence processing. Instead of undoing the Hadamard transform during the forward pass, we adopt the computational invariance theorem from SliceGPT [Ashkboos et al., 2024] to fuse the transformations into the weights where possible. Instead of requiring two Hadamard transforms per weight-matrix in the forward pass, QuaRot requires just $1\frac{1}{2}$ Hadamard transforms per transformer layer. Computational invariance also means that the *activations* are incoherence-processed, enabling them to be effectively quantized. We also apply a similar technique to the attention block and quantize the KV cache in 4 bits with minimal accuracy loss.

## 3 Background

Here we introduce some mathematical concepts and notation that are necessary for QuaRot.

### 3.1 Orthogonal, Rotation and Hadamard Matrices

An orthogonal matrix $\mathbf{Q}$ is a square matrix such that $\mathbf{Q}\mathbf{Q}^\top = \mathbf{I}$. In this work, we consider only real orthogonal matrices. A rotation matrix is an orthogonal matrix. A Hadamard matrix is an orthogonal matrix with entries drawing from $\{+1,-1\}$. A Walsh-Hadamard matrix is a square matrix of size $d = 2^n$, with

$$\mathbf{H}_2 = \frac{1}{\sqrt{2}} \left[ \begin{array}{cc} 1 & 1 \\ 1 & -1 \end{array} \right] \qquad \text{and} \qquad \mathbf{H}_{2^n} = \mathbf{H}_2 \otimes \mathbf{H}_{2^{n-1}} \,. \tag{1}$$

These identities give rise to the Walsh-Hadamard transform, which computes the matrix-vector product $\mathbf{H}\boldsymbol{x}$ in $\mathcal{O}(d \log_2(d))$ operations.

For matrix sizes that are not $2^n$, the existence of a Hadamard matrix is not guaranteed. A useful list of known Hadamard matrices is made available by Sloane [2024]. Where we require a Hadamard matrix of size $d \neq 2^n$, we factorize $d = 2^n m$, where $m$ is the size of a known Hadamard matrix. Then we use a Kronecker construction $\mathbf{H}_d = \mathbf{H}_{2^n} \otimes \mathbf{H}_m$. This allows computation of $\mathbf{H}_d \boldsymbol{x}$ in $\mathcal{O}(d(m+n))$ operations.

Following Tseng et al. [2024] we make use of *randomized* Hadamard matrices where convenient. Let $\boldsymbol{s}$ be a vector containing random draws from $\{+1,-1\}$, and $\tilde{\mathbf{H}} = \mathbf{H} \operatorname{diag}(\boldsymbol{s})$. It is straightforward to see that $\tilde{\mathbf{H}}$ is also an orthogonal matrix.

### 3.2 Incoherence Processing

The idea of *incoherence processing* was introduced by [Chee et al., 2024] in the context of weight normalization for weight-only LLM quantization. We define a weight matrix $\mathbf{W}$ to be $\mu$-incoherent if

$$\max(\mathbf{W}) \leq \mu \|\mathbf{W}\|_F / \sqrt{mn} \tag{2}$$

where max is the element-wise max of the matrix, and $mn$ is the number of elements. A weight matrix that has high incoherence is hard to quantize: the largest element is an outlier relative to the magnitude of the average element. Chee et al. [2024] showed that multiplying a weight matrix on the left and right by an orthogonal matrix can reduce the incoherence, making matrices easier to quantize. In this work we adopt a similar technique, multiplying weight matrices by orthogonal matrices to improve incoherence, though we add fewer operations to the forward pass. Importantly, we additionally apply incoherence processing to the activations, enabling improved weight and activation quantization. Figure 1 shows the effect of applying incoherence processing to the activations of LLAMA-2 .

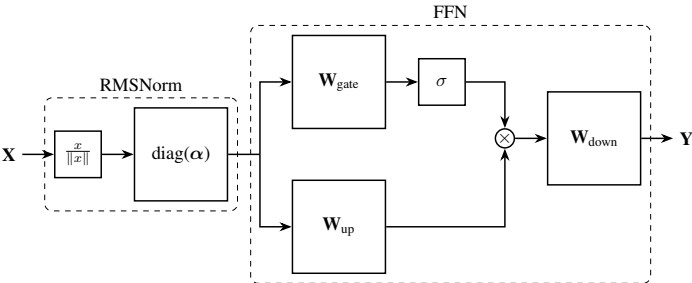

Figure 2: The gated feed-forward network used in most LMs, including the pre-positioned RMSNorm. The input signal is divided by its norm, and re-scaled by parameters $\alpha$. Two linear blocks, $\mathbf{W}_{\text{up}}$ and $\mathbf{W}_{\text{gate}}$ are applied. The activation function $\sigma$ is applied to the gated signal, and the two signals are element-wise multiplied together. The final linear block $\mathbf{W}_{\text{down}}$ produces the output signal $\mathbf{Y}$. Before quantization, different operations are performed either in single (32 bit) or half (16 bit) precision.

## 3.3 Transformer structures

Large Language Models are neural networks with repeating attention and feed-forward layers. We introduce our notation through Figures 2 and 5, which show the construction of these blocks. We assume that the construction of the network is "pre-norm", in that each block is preceded by a LayerNorm or RMSNorm operation. We also assume that the feed-forward network uses a gated architecture, as in LLAMA-2 , though our methodology is straightforwardly applied to MLP architectures also.

## 3.4 Computational Invariance

The computational invariance theorem [Ashkboos et al., 2024, Theorem 1] states that the weights and between-block activations in a transformer can be transformed using an orthogonal matrix with no change to the model output. Here we sketch the main idea. If $\mathbf{W}_{\text{in}}$ is a weight matrix that appears on the left of a transformer block (i.e., $\mathbf{W}_{\text{gate}}, \mathbf{W}_{\text{up}}$ in Figure 2, or $\mathbf{W}_k, \mathbf{W}_q, \mathbf{W}_v$ in Figure 5) then we can multiply on the left by an orthogonal matrix $\mathbf{Q}$, and cancel out this effect by multiplying the output matrix ($\mathbf{W}_{\text{down}}, \mathbf{W}_{\text{out}}$) by $\mathbf{Q}^\top$. This applies despite the fact that RMSNorm is applied between the two blocks, so long as no re-scaling happens in the RMSNorm block (and in practice, we absorb any re-scaling into adjacent weight matrices first). Conceptually, this is because RMSNorm divides the activations by their norm, and applying a rotation $\mathbf{Q}$ to the activations does not affect the norm. We have the commutation property

$$\text{RMSNorm}(\mathbf{X}) = \text{RMSNorm}(\mathbf{X}\mathbf{Q}^\top)\mathbf{Q}, \tag{3}$$

where we assume here that RMSNorm applied to each row of the activations $\mathbf{X}$ as $\boldsymbol{x}_i \leftarrow \boldsymbol{x}_i / \|\boldsymbol{x}_i\|$. This means that multiplying an output matrix by $\mathbf{Q}^\top$ makes the linear layer output $\mathbf{X}\mathbf{Q}^\top$, which is normalized and then passed into the next block whose input weight matrix is now $\mathbf{Q}\mathbf{W}$, and so *this* linear layer outputs the original activations without modification.

## 4  Method

QuaRot consists of two stages. In the first stage, the model weights are manipulated (in full precision), and two additional Hadamard operations are inserted into the model's forward pass. In the second stage, the weights are quantized using some existing method, and quantization operations are added to the forward pass to enable on-line quantization of the activations (and caches). By default, we use GPTQ [Frantar et al., 2022] for quantizing weights, whilst activations are quantized on-the-fly using a simple round-to-nearest scheme. Figures 3 and 6 show updated block diagrams for the forward pass with QuaRot modifications, including updated weight matrices, inserted blocks and the bit-width of weights and activations.

**Stage 1a: Weight Modification.**   We first make use of computational invariance to multiply each weight matrix by an orthogonal matrix. To enable this, the linear parts of LayerNorm or RMSNorm are fused into adjacent weight matrices. Figure 3 shows how the feed-forward block of a transformer is modified by removing the scaling operation from RMSNorm ($\text{diag}(\boldsymbol{\alpha})$) and absorbing into the

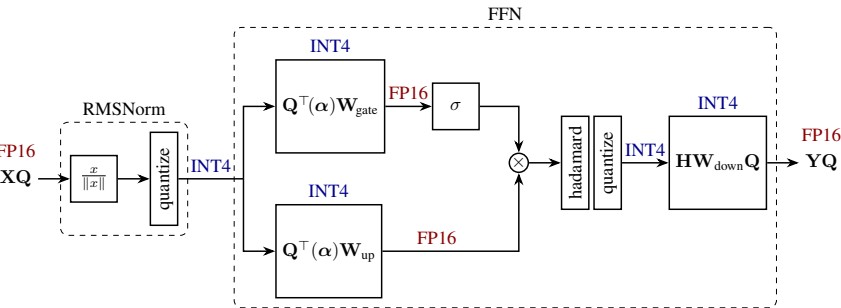

Figure 3: QuaRot applied to a LLaMa-style FFN. The RMSNorm scaling ($\alpha$) has been absorbed into the weight matrices (($\alpha$) is a diagonal matrix with RMSNorm parameters). The hidden state $\mathbf{X}$ has been rotated by $\mathbf{Q}$, which is canceled out by the absorption of $\mathbf{Q}^\top$ into the first two weight matrices. All weights are stored in INT4, and all activations immediately before the weights are also quantized to INT4. The result of the matmul between the INT4 weights and activations on a TensorCore is INT32, which we immediately cast (and scale) to FP16 which is the default precision of the model. Whilst the signal is still in FP16, we perform a single on-the-fly Hadamard transform before quantizing and computing a (modified) down-proj, which results in a rotated output $\mathbf{YQ}$.

subsequent weight matrices. We select a randomized Hadamard matrix with size that matches the hidden dimension of the model and pre- or post-multiply each weight matrix. In Figures 3 and 6 this matrix is denoted $\mathbf{Q}$. For example the key-projection weight matrix $\mathbf{W}_k$ is modified as

$$\mathbf{W}_k \leftarrow \mathbf{Q}^\top \mathrm{diag}(\boldsymbol{\alpha})\mathbf{W}_k \,, \tag{4}$$

and similarly for other weight matrices. Matrices that appear on the *output* side of a block are post-multipled by $\mathbf{Q}$.

This weight modification does not affect the output of the model (assuming sufficient precision) as per the computational invariance theorem [Ashkboos et al., 2024]. We note that the modified weights resemble the modifications used in QuIP# [Tseng et al., 2024], reducing the incoherence of the weights, though our modification does not require any additional processing at run-time. Additionally, the activation matrix passed between blocks of the transformer is also incoherence processed, becoming $\mathbf{X} \leftarrow \mathbf{XQ}$. Figure 1 shows the result of this processing: we see that the processed activations no longer contain any outliers.

**Stage 1b: Rotate FFN activations.** With the above weight-modifications in place, we have multiplied many weight matrices on one side by a Hadamard matrix and the activations have been changed. It remains to improve the quantization of the activations *within* each block, which we achieve by inserting on-line Hadamard operations.

We first insert a Hadamard operation into the feed-forward network, before the down-projection matrix. This operation is performed in full precision, and implemented using a fast kernel following Tseng et al. [2024]. This operation is implicitly reversed by fusing a Hadamard matrix into the down-projection matrix of the network: $\mathbf{W}_{\mathrm{down}} \leftarrow \mathbf{HW}_{\mathrm{down}}$. Combined with the global matrix $\mathbf{Q}$, this means that the down-projection matrix now becomes $\mathbf{HW}_{\mathrm{down}}\mathbf{Q}$ (see Figure 3).

**Stage 1c: Attention Value Projection.** Next, we apply an additional Hadamard operation to each attention block. This modification is partially on-line, and partially fused into the weight matrices as we will now detail.

First, note that in the computation of attention, the $\mathbf{W}_v$ and $\mathbf{W}_{\mathrm{out}}$ matrices are implicitly multiplied together within each head. To see this, note that the attention computation consists of

$$\mathbf{Y} = \mathrm{concat}[(\mathbf{P}_1\mathbf{V}_1)\dots(\mathbf{P}_{n_h}\mathbf{V}_{n_h})]\mathbf{W}_{\mathrm{out}} \tag{5}$$

$$= \sum_{h=1}^{H} \mathbf{P}_h\mathbf{X}\mathbf{W}_v^{(h)}\mathbf{W}_{\mathrm{out}}^{(h)} \tag{6}$$

where $\mathbf{P}_h$ is a sequence-length sized square matrix computed by softmaxing keys and values, and $\mathbf{V}_h = \mathbf{XW}_v^{(h)}$ is the value matrix for one head. This presents an opportunity to perform additional

processing on $\mathbf{W}_v$ and $\mathbf{W}_{\text{out}}$ using a Hadamard matrix $\mathbf{H}_{d_h}$ which matches the dimension of each head:

$$\mathbf{W}_v^{(h)} \leftarrow \mathbf{W}_v^{(h)}\mathbf{H}_{d_h}, \qquad \mathbf{W}_{\text{out}}^{(h)} \leftarrow \mathbf{H}_{d_h}\mathbf{W}_{\text{out}}^{(h)}. \tag{7}$$

Substituting these modifications into equation (6), we see that the computed result of attention remains unchanged. Since the weights for each head are concatenated in the weight representation, we can equivalently perform a single Kronecker structured multiplication:

$$\mathbf{W}_v \leftarrow \mathbf{W}_v(\mathbf{I} \otimes \mathbf{H}_{d_h}), \qquad \mathbf{W}_{\text{out}} \leftarrow (\mathbf{I} \otimes \mathbf{H}_{d_h})\mathbf{W}_{\text{out}}. \tag{8}$$

This transformation has now been applied head-wise to the weight matrices, and results in computed activations (emitted by the block *multi-head attention*) rotated head-wise also. To complete a "full" Hadamard operation on the attention-activations, sharing the transform across heads, we make use of the identity

$$\mathbf{H}_{n_h \times d_h} = (\mathbf{I} \otimes \mathbf{H}_{d_h})(\mathbf{H}_{n_h} \otimes \mathbf{I}) \tag{9}$$

which holds when the number of heads $n_h$ and the dimension of each head $d_h$ are both powers of 2. Since we have already applied $(\mathbf{I} \otimes \mathbf{H}_{d_h})$ to both $\mathbf{W}_v$ and $\mathbf{W}_{\text{out}}$, it remains to apply $(\mathbf{H}_{d_h} \otimes \mathbf{I})$ to $\mathbf{W}_{\text{out}}$, which results in a complete transformation of $\mathbf{W}_{\text{out}} \leftarrow \mathbf{H}\mathbf{W}_{\text{out}}$, and to insert a block into the forward pass that computes $\mathbf{Z} \leftarrow \mathbf{Z}(\mathbf{H}_{n_h} \otimes \mathbf{I})$ where $\mathbf{Z}$ is the attention activation. This block is denoted *Hadamard heads* in Figure 6 and can be computed efficiently using a reshape to deal with the Kronecker structure, and a Walsh-Hadamard transform on the reshaped data.

**Stage 1d: Key Rotation.**  Using the method above, we can successfully quantize the value vectors. However, key vectors in the attention module are also known to suffer from outliers [Hooper et al., 2024, Liu et al., 2024]. Similar to above, we can use a Hadamard rotation to alleviate this issue, allowing us to have a fully quantized KV cache. First note that the attention scores $\mathbf{P}_1, \ldots, \mathbf{P}_h$ are computed as:

$$\mathbf{Q} \leftarrow \text{Pos}(\mathbf{X}\mathbf{W}_q) = \text{concat}[\text{Pos}(\mathbf{Q}_1), \ldots, \text{Pos}(\mathbf{Q}_{n_h})] \tag{10}$$

$$\mathbf{K} \leftarrow \text{Pos}(\mathbf{X}\mathbf{W}_k) = \text{concat}[\text{Pos}(\mathbf{K}_1), \ldots, \text{Pos}(\mathbf{K}_{n_h})] \tag{11}$$

$$\mathbf{P}_h \leftarrow \text{Softmax}(\alpha \, \text{Pos}(\mathbf{Q}_h) \, \text{Pos}(\mathbf{K}_h^\top) \odot \mathbf{M}), \tag{12}$$

where $\alpha$ is the Softmax scale usually set to $\frac{1}{\sqrt{d_h}}$, $\mathbf{M}$ is the attention mask (e.g., causal), and $\text{Pos}$ denotes the positional embedding. Previously, positional embedding was only added before the first layer to the input, in which case $\text{Pos}$ is an identity function. However, recent methods such as RoPE [Su et al., 2021] add position information directly to the key and query vectors.

We can now observe the same interaction between $\mathbf{Q}$ and $\mathbf{K}$ as we observed between $\mathbf{W}_v$ and $\mathbf{W}_{\text{out}}$. However, the existence of $\text{Pos}$ prevents us from directly fusing the Hadamard matrix into $\mathbf{W}_q$ and $\mathbf{W}_k$. Therefore, we use online head-wise Hadamard rotation to rotate both the queries and keys. As a result, the computation of query and key matrices is altered as follows:

$$\mathbf{Q} \leftarrow \text{Pos}(\mathbf{X}\mathbf{W}_q)(\mathbf{I} \otimes \mathbf{H}_{d_h}) = \text{concat}[\text{Pos}(\mathbf{Q}_1)\mathbf{H}_{d_h}, \ldots, \text{Pos}(\mathbf{Q}_{n_h})\mathbf{H}_{d_h}] \tag{13}$$

$$\mathbf{K} \leftarrow \text{Pos}(\mathbf{X}\mathbf{W}_k)(\mathbf{I} \otimes \mathbf{H}_{d_h}) = \text{concat}[\text{Pos}(\mathbf{K}_1)\mathbf{H}_{d_h}, \ldots, \text{Pos}(\mathbf{K}_{n_h})\mathbf{H}_{d_h}]. \tag{14}$$

Since both queries and keys are rotated, the final attention scores $\mathbf{P}_1, \ldots, \mathbf{P}_h$ remain unchanged. We note that an alternative to the above process is caching the keys before applying the positional encoding. This approach (called Pre-RoPE Caching [Hooper et al., 2024]) needs the inverse rotation to be applied online before applying the positional encoding but removes the need to rotate the query vector. It also adds the overhead of rotating the keys and values for every query. Given that at the time of decoding there is a single query vector and many cached key vectors, we use Post-RoPE caching. This helps us to apply a Hadamard transformation on a single token at each decoding step.

Overall, our modifications to the forward pass, including the insertion of special Hadamard blocks and adjustments to the weights do not change the forward pass of the model. The effect is that the activations between blocks have been multiplied by a Hadamard matrix, and the activations within blocks are processed on-line using Hadamard transforms in a way that is undone by corresponding weight matrix modifications. We are now ready to quantize the weights and activations.

**Stage 2a: Weight Quantization.**  We apply GPTQ [Frantar et al., 2022] to quantize the weights of the network. We note that after the above forward-pass modifications, any quantization method could be applied. In subsequent sections, we show that a simple round-to-nearest (RTN) scheme can be applied instead of GPTQ, at the cost of some accuracy.

**Stage 2b: Online Quantization Operations.** With the weights quantized, we are ready to apply operations to the forward pass that quantize the activations. Following PyTorch implementation, we leave the computation of RMSNorm (without scaling) in FP32. We quantize the input of the linear layers using symmetric per-token (rows of the input matrix). During symmetric quantization, the row scales are computed by dividing the maximum absolute value of each token by 7 (largest representable number in INT4). We then divide each row to its corresponding scale and round the result to its nearest integer. The dequantization is also done by casting the INT32 output of GEMM into FP16, multiply the corresponding scale for the row (from input scales) and column (from weight scales).

**Stage 2c: Quantized Attention.** Attention is significantly memory bound for longer sequences and larger batch sizes. Having rotated both keys and values, we can successfully quantize the cache into low bit-width. This reduces the number of IO operations needed. We keep the queries in FP16 and use online softmax calculation similar to Flash Attention [Dao et al., 2022]. After a segment of the KV vectors are loaded from the memory, we dequantize and compute the dot product in FP16.

## 5 Experimental Validation

**Setup.** We implement QuaRot using Hugging Face [Wolf et al., 2019] on top of the PyTorch framework [Paszke et al., 2019]. To quantize the inputs, we use per-token symmetric quantization (a single scale for every row) with a constant clipping ratio of 0.9 in all our experiments. We quantize the KV caches using asymmetric quantization with a group size 128 with a constant clipping ratio of 0.95. For weight quantization, we use round-to-nearest (RTN) and GPTQ [Frantar et al., 2022] with per-column (also known as per-channel) symmetric quantization, where we extract the clipping ratio using a linear search over the squared error. We use 128 samples from WikiText-2 [Merity et al., 2016] training set with 2048 sequence length as the calibration set during GPTQ quantization. On a single NVIDIA A100 GPU, modifying LLAMA2-70B with QuaRot takes 5 minutes and quantizing the model with GPTQ takes a further 2 hours. We present LLAMA-3 results in Appendix A.8.

**Models, Tasks, and GPUs.** We evaluate QuaRot on the LLAMA-2 family [Touvron et al., 2023] on both language generation and zero-shot tasks. We implement our low-level CUDA kernel to perform 4-bit matrix-multiplication using the CUTLASS [NVIDIA, 2023] library. We use the FlashInfer [Ye, 2023] library for implementing our KV cache quantization. As we target consumer-type GPUs, we evaluate all the performance experiments on NVIDIA RTX 3090 GPUs.

### 5.1 Accuracy Results

**Language Generation Tasks.** First, we evaluate the accuracy of QuaRot on the language generation task. Table 1 shows the perplexity of LLAMA-2 models on WikiText-2 when we quantize the weights using GPTQ. We compare against 4-bit SmoothQuant [Xiao et al., 2023] and OmniQuant [Shao et al., 2023]. We also include the QUIK [Ashkboos et al., 2023] results when they keep all the layers (including down-projection) in 4 bits. QuaRot outperforms all previous work with at most 0.63 perplexity loss (0.47 on LLAMA2-70B model) without any re-training (as in OmniQuant) nor higher precision outlier features and asymmetric quantization (as in QUIK). We also apply group-wise quantization to compare against Atom [Zhao et al., 2023] on the same number of groups for weight and activations. In this setting, QuaRot doesn't need to keep any higher precision features and related operations (like re-ordering). QuaRot outperforms Atom with 0.1 perplexity points in the 7B model. On the 13B model, we get the same perplexity number as Atom.

**Zero-Shot Tasks.** Next, we focus on evaluating QuaRot on six important zero-shot tasks: PIQA [Bisk et al., 2020], WinoGrande [Sakaguchi et al., 2021], HellaSwag [Zellers et al., 2019], LAMBADA (OpenAI) [Radford et al., 2019], and Arc (Easy and Challenge) [Clark et al., 2018]. We use the LM Evaluation Harness [Gao et al., 2021] with default parameters for our experiments. Table 2 shows the accuracy of our scheme on the above tasks as well as the average score. On LLAMA-2 family, QuaRot preserves the accuracy with at most 4.18% average score loss (1.09% for 70B model).

### 5.2 Performance Analysis

We implement QuaRot using CUDA/12.1 on top of PyTorch and use CUTLASS for performing INT-4 matrix multiplication on TensorCore (where the results will be saved in an INT32 accumulator). In this section, we evaluate the performance of our kernels for both prefill and decoding steps on NVIDIA RTX 3090 GPU. We provide all our experiments on a single transformer block as the whole

Table 1: WikiText-2 perplexity results on 4-bit quantization of LLAMA-2 models with 2048 sequence length. We extract the results for SmoothQuant and OmniQuant results of [Shao et al., 2023]. 128G shows the group-wise quantization with group size 128.Here, we quantize all weights, activations, and caches in 4-bits in QuaRot.

| Method | Weight Quantization | #Outlier Features | LLAMA-2 | | |
|---|---|---|---|---|---|
| | | | 7B | 13B | 70B |
| Baseline | - | - | 5.47 | 4.88 | 3.32 |
| SmoothQuant | RTN | 0 | 83.12 | 35.88 | - |
| OmniQuant | RTN | 0 | 14.26 | 12.30 | - |
| QUIK-4B | GPTQ | 256 | 8.87 | 7.78 | 6.91 |
| QuaRot | GPTQ | 0 | **6.10** | **5.40** | **3.79** |
| Atom-128G | GPTQ-128G | 128 | 6.03 | **5.26** | - |
| QuaRot-128G | | 0 | **5.93** | **5.26** | **3.61** |

Table 2: Zero-shot accuracy of LLAMA-2 models with 4-bit (A4W4KV4) QuaRot on PIQA (PQ), WinoGrande (WG), HellaSwag (HS), Arc-Easy (A-e), Arc-Challenge (A-c), and LAMBADA (LA).

| Model | Method | PQ | WG | HS | A-e | A-c | LA | Avg. |
|---|---|---|---|---|---|---|---|---|
| LLAMA2-7B | FP16 | 79.11 | 69.06 | 75.99 | 74.58 | 46.25 | 73.90 | 69.82 |
| | QuaRot | 76.77 | 63.77 | 72.16 | 69.87 | 40.87 | 70.39 | 65.64 |
| LLAMA2-13B | FP16 | 80.47 | 72.22 | 79.39 | 77.48 | 49.23 | 76.75 | 72.59 |
| | QuaRot | 78.89 | 70.24 | 76.37 | 72.98 | 46.59 | 73.67 | 69.79 |
| LLAMA2-70B | FP16 | 82.70 | 77.98 | 83.84 | 80.98 | 57.34 | 79.58 | 77.07 |
| | QuaRot | 82.43 | 76.24 | 81.82 | 80.43 | 56.23 | 78.73 | 75.98 |

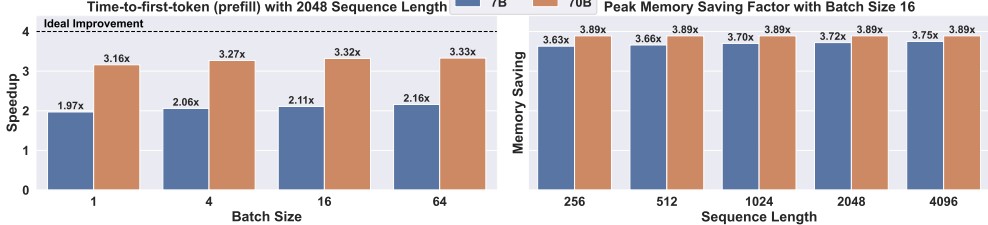

Figure 4: Performance of the QuaRot kernel on a single transformer block of LLAMA-2 models using NVIDIA RTX 3090 GPU. **Left**: For the speedup results, we evaluate using sequence length 2048 with different batch sizes. **Right**: Peak memory saving during decoding of 50 tokens with different prefill sequence lengths using batch size 16.

model does not fit on our GPU cluster for large batch sizes. We provide more performance analysis of our kernels (as well as complete results) in Appendix A.10.

**Prefill Stage Performance Increases.** For the compute-bound prefill stage, we present the speedups of using QuaRot on 2048 sequence length with different batch sizes in Figure 4 **Left**. On LLAMA2-7B model, we get 1.97x-2.16x speedup over the FP16 implementation using our QuaRot kernel. The speedup increases with batch sizes as the computation will become a bottleneck in larger batch sizes. on LLAMA2-70B model, we get up to 3.33x speedup. Note that our performance results could be improved by optimizing our kernels (e.g., fusing the quantization operations into the MatMul).

**Decoding Stages Memory Saving.** Finally, we evaluate the memory improvement which is the main bottleneck of the decoding stage. Figure 4 **Right** shows the peak memory saving on LLAMA-2 models. We provide results for LLAMA2-7B and LLAMA2-70B models. In both models, we get at least 3.63x peak memory saving compared to FP16 case during the decoding stage. Note that the KV cache is larger in LLAMA2-7B model as the LLAMA2-70B uses grouped-query attention [Ainslie et al., 2023]. In the LLAMA2-7B model, the memory saving increases with the sequence length, resulting in up to 3.75x memory saving. on LLAMA2-70B model, we get 3.89x savings in almost all the cases. We expect these values to be larger for the whole model (instead of just the single layer

Table 3: WikiText-2 Perplexity and zero-shot accuracy of QuaRot on the LLAMA-2 family using 4-
and 8-bits with Round-to-Nearest (RTN) weights and activation quantization. For zero-shot tasks, we
use PIQA (PQ), WinoGrande (WG), HellaSwag (HS), Arc-Easy (A-e), Arc-Challenge (A-c), and
LAMBADA (LA). We quantize all weights, activations, and caches.

| Model | Method | Precision | PPL ↓ | PQ ↑ | WG ↑ | HS ↑ | A-e ↑ | A-c ↑ | LA ↑ | Avg. ↑ |
|-------|--------|-----------|-------|------|------|------|-------|-------|------|--------|
| 7B | Baseline | FP16 | 5.47 | 79.11 | 69.06 | 75.99 | 74.58 | 46.25 | 73.90 | 69.82 |
| | QuaRot-RTN | INT4 | 8.37 | 72.09 | 60.69 | 65.40 | 58.88 | 35.24 | 57.27 | 58.26 |
| | | INT8 | 5.50 | 78.94 | 68.67 | 75.80 | 74.79 | 45.39 | 74.33 | 69.65 |
| 70B | Baseline | FP16 | 3.32 | 82.70 | 77.98 | 83.84 | 80.98 | 57.34 | 79.58 | 77.07 |
| | QuaRot-RTN | INT4 | 4.14 | 80.69 | 75.14 | 79.63 | 77.57 | 51.71 | 77.02 | 73.63 |
| | | INT8 | 3.33 | 82.97 | 77.98 | 83.67 | 80.77 | 58.11 | 79.53 | 77.17 |

Table 4: WikiText-2 perplexity of 4-bit QuaRot with various group-sizes on LLAMA-2 models. We
use GPTQ during the weight quantization. In all cases, we keep the KV cache group-size to 128
(same as the head dimension). 128G shows the group-wise quantization with 128 group size.

| Method | LLAMA-2 | | |
|--------|---------|---------|---------|
| | 7B | 13B | 70B |
| Baseline | 5.47 | 4.88 | 3.32 |
| QuaRot | 6.10 | 5.40 | 3.79 |
| QuaRot-256G | 5.98 | 5.28 | 3.63 |
| QuaRot-128G | 5.93 | 5.26 | 3.61 |
| QuaRot-64G | 5.88 | 5.25 | 3.58 |

here) since as the number of layers increases the effect of constant size objects in memory becomes
much less significant.

## 5.3 Ablation Studies

To evaluate different aspects of QuaRot, we evaluate the use of **Round-to-Nearest Weight Quanti-
zation**, **Group-wise Quantization** (with different group sizes), and **KV cache Quantization** with
different bit-width combinations (Appendix A.3). In addition, we investigate the role of applying
Hadamard transformation on the **Weight-only Quantization** schemes (Appendix A.4) as well as
using **Random Orthogonal Matrices** (Appendix A.5) instead of Hadamard matrices. Finally, we
evaluate the accuracy of our quantized models when we apply **FP16 Hadamard Transformation**
(Appendix A.7).

**Round-to-Nearest Weight Quantization.** GPTQ is our default choice for weight quantization in
QuaRot. Here, we study the role of quantizing the weights using Round-to-Nearest (RTN). Table 3
shows that applying RTN weight quantization fully maintains the FP16 model accuracy in 8 bits.
We note that RTN does not need any calibration set or hyper-parameter during the quantization.
Comparing Table 3 and 2, we conclude that in 4 bits, the gap between QuaRot-RTN and QuaRot-
GPTQ decreases when the model size is increased (2.27 on LLAMA2-7B and 0.34 on LLAMA2-70B )
showing that GPTQ is a better option in smaller models. For more detailed results see Appendix A.6.

**Group-wise Quantization.** Table 4 shows the accuracy of applying QuaRot with various
group-sizes for the activations and weights. The results show a clear trade-off between the accuracy
and the group-sizes: smaller group-sizes give better accuracy (but require more bits to store scales for
each group and more complex matrix-multiplication kernels).

## 6 Conclusion

We introduce QuaRot: a method which uses Hadamard matrices to eliminate outliers in the activations
and KV cache of pre-trained LLMs, enabling end-to-end 4-bit quantization for the first time (to
the best of our knowledge). Quantizing LLAMA2-70B to 4 bits with QuaRot maintains 99% of the

downstream task performance of the FP16 baseline, with a 2.16× speedup on RTX 3090 GPUs during the prefill stage (and up to 3.39× memory saving during the decoding stage). Quantizing all LLAMA-2 models to 6 and 8 bits is lossless.

Opportunities to build on QuaRot include quantizing the residuals and extending the method to mixture-of-experts architectures. In terms of hardware, end-to-end INT4 inference with QuaRot could be exploited to give similar speedups as that of the recently announced NVIDIA B200 GPU architecture, while being much cheaper to implement compared to the floating point (FP4) format.

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

# A Appendix

## A.1 QuaRot on the Attention Module

Figure 5 shows the original attention module in large language models with RoPE. The input of the attention module is already rotated using the randomized Hadamard matrix $\mathbf{Q}$ (see Section 4) and in the first step, we fuse the inverse of such matrices into the input linear layers of the attention. In the next step, we fuse the exact Hadamard matrices on each block of the columns (proportional to each head) on the `V_projection` layer to make sure that the Values will be rotated at the output of that layer. In the next step, we apply exact Hadamard transformations on the Keys and Queries and quantize the KV after RoPE operation (note that the Keys and Queries Hadmard transformations will be canceled during the attention operation). Finally, we apply another Hadamard transformation between heads before `Out_projection` layer and fuse the inverse into the weights. Figure 6 shows the result of applying QuaRot on the attention module.

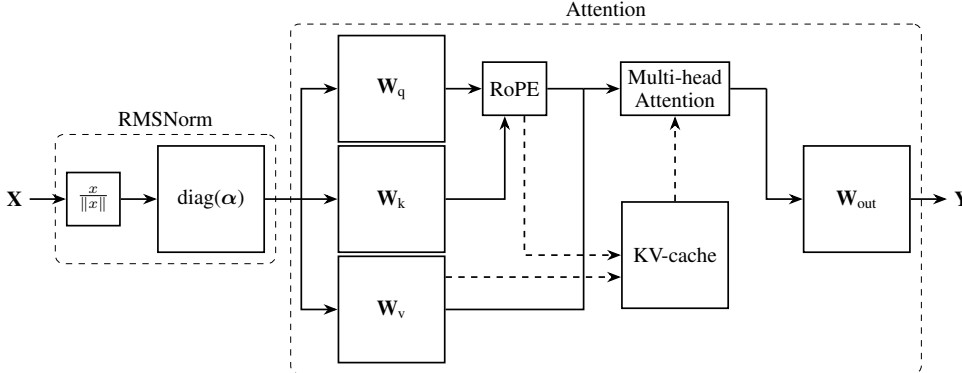

Figure 5: Flow diagram of a self-attention block as used in most LMs, including the pre-positioned RMSNorm. Solid arrows represent flow during training, prefill and inference of each token. Dashed arrows show access to and from the KV cache, used at generation-time. The RoPE block computes relative positional embeddings.

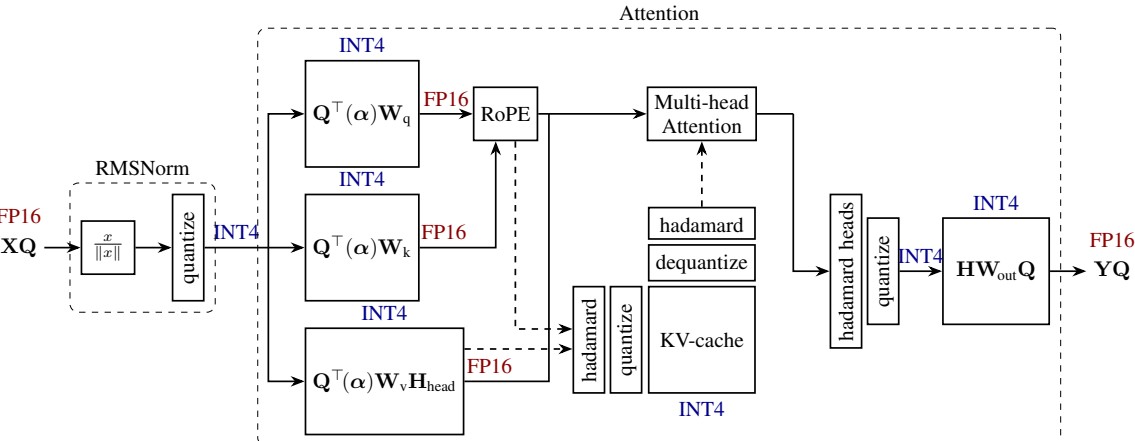

Figure 6: QuaRot applied to an attention component. The RMSNorm scaling $\boldsymbol{\alpha}$ is absorbed into the input weight matrices, and the hidden state has been rotated by $\mathbf{Q}$ in the same way as for the FFN block (see previous figure). Colored labels show the bit-width of each flow, and dashed lines show the flow to/from the KV cache.

## A.2 Clipping Ratio Ablation

We use the clipping ratio for both weights and activations during the quantization. During the weight quantization, we apply a linear search over the MSE error to extract the best clipping ratio for each

column of the weight matrix. However, this is not possible as we quantize the inputs on the fly during the inference and we need to use a constant clipping ratio for such quantization. We conclude that using 0.95 and 0.9 are suitable during asymmetric (KV cache) and symmetric (inputs) quantization which matches the finding from [Zhao et al., 2023].

Table 5: WikiText perplexity of LLAMA2-7B with different clipping ratio. To study the effect of various clipping ratios, we keep the rest of the model in full precision.

|  | 1.0 | 0.95 | 0.9 | 0.85 |
|---|---|---|---|---|
| Input Quantization | 5.938 | 5.910 | **5.828** | 5.850 |
| KV Cache Quantization | 5.513 | **5.510** | 5.517 | 5.532 |

## A.3 KV Cache Quantization Ablation

We keep the rest of the model (including weights and activations) in high precision and apply our group-wise asymmetric quantization (with group-size 128) with various precision to keys and values. Table 6 shows the results of using various precision during KV cache quantization. The results show a negligible (at most 0.21) perplexity degradation up to 3-bit KV cache (0.07 for LLAMA2-70B model). In addition, by comparing the 3 and 4-bit quantization, we can see that compared to the values, keys are more sensitive to quantization as keeping the keys in 4-bits and values in 3-bits has 0.03 perplexity loss (0.18 for 3-bit keys and 4-bit values) on the LLAMA2-7B model. This matches the previous study on KV cache quantization [Hooper et al., 2024, Liu et al., 2024]. The results show that using 3-bit KV-caches results in a better accuracy (5.68 on LLAMA2-7B model) compared to keeping the keys in 4-bits and quantizing the values using 2-bits (with 5.75 perplexity on LLAMA2-7B model).

Table 6: WikiText-2 perplexity with various KV cache precision using QuaRot.

| K bits | V bits | LLAMA-2 | | |
|---|---|---|---|---|
| | | 7B | 13B | 70B |
| 16 | 16 | 5.47 | 4.88 | 3.32 |
| 4 | 4 | 5.51 | 4.91 | 3.33 |
| 4 | 3 | 5.54 | 4.93 | 3.35 |
| 4 | 2 | 5.75 | 5.09 | 3.43 |
| 3 | 4 | 5.65 | 5.01 | 3.38 |
| 3 | 3 | 5.68 | 5.02 | 3.39 |
| 3 | 2 | 5.93 | 5.21 | 3.48 |
| 2 | 4 | 8.06 | 6.42 | 3.89 |
| 2 | 3 | 8.18 | 6.50 | 3.92 |
| 2 | 2 | 9.23 | 7.07 | 4.13 |

## A.4 Weight-only Quantization Ablation

QuaRot improves the quality of quantized models by removing the outlier features during the Hadamard transformations. As we fuse the Hadamard matrices into the weights, we study the role of these transformations for weight-only quantization (we keep the rest of the data-types in FP16). Table 7 shows the WikiText-2 perplexity results with asymmetric quantization. Using GPTQ quantization, QuaRot improves the perplexity by up to 2.65 in 4 bits. In addition, applying QuaRot improves the quality more in lower precision (2-3 bits) in all models. QuaRot also improves the RTN quantization up to 0.24 perplexity points. GPTQ still has a lower perplexity in 2-3 bits. However, applying QuaRot improves the quality of GPTQ in 2 bits to a non-trivial value (5.6 on the LLAMA2-70B model).

Table 7: Weight-only quantization results on WikiText-2 on LLAMA-2 models. We use asymmetric per-column quantization and keep the inputs and KV cache in FP16. We show the perplexity results >100 by Inf. We show the failed GPTQ experiments using NaN.

| Method | LLAMA-2 | | | | | | | | |
| | 7B | | | 13B | | | 70B | | |
| | 5.47 | | | 4.88 | | | 3.32 | | |
|---|---|---|---|---|---|---|---|---|---|
| Baseline | 5.47 | | | 4.88 | | | 3.32 | | |
| | A16W4 | A16W3 | A16W2 | A16W4 | A16W3 | A16W2 | A16W4 | A16W3 | A16W2 |
| RTN | 6.99 | Inf | Inf | 6.32 | Inf | Inf | 4.45 | 42.11 | Inf |
| GPTQ | 8.25 | NaN | NaN | 5.65 | 9.51 | Inf | 3.87 | 5.91 | 25.30 |
| QuaRot-RTN | 6.76 | Inf | Inf | 5.48 | 48.89 | Inf | 3.66 | 5.25 | Inf |
| QuaRot-GPTQ | 5.60 | 6.09 | 22.07 | 5.00 | 5.37 | 10.41 | 3.41 | 3.72 | 5.60 |

## A.5 Random Orthogonal Matrices Ablation

QuaRot fuses Hadamard transformations into weight matrices to eliminate outliers. However, due to the computational invariance property in LLMs, any orthogonal matrix can be fused to the model and we only need to apply an online $1\frac{1}{2}$ Hadamard transformations in each layer (see Section 4). Here, we study the use of random orthogonal matrices in QuaRot. We start with a uniformly random matrix and apply QR decomposition to make it orthogonal before fusing it into the weights.

Table 8: WikiText-2 perplexity of 4-bit QuaRot on LLAMA-2 models with different orthogonal matrices.

| Method | LLAMA-2 | | |
| | 7B | 13B | 70B |
|---|---|---|---|
| Baseline | 5.47 | 4.88 | 3.32 |
| QuaRot (Random) | 7.45 | 5.84 | 4.07 |
| QuaRot (Hadamard) | 6.10 | 5.40 | 3.79 |

Table 8 shows the results of applying random orthogonal matrices on LLAMA-2 models. Random orthogonal matrices are not as good as random Hadamard transformations and we have up 1.35 perplexity gap on LLAMA2-7B . However, as the model size increases, the gap decreases, resulting in a perplexity change of 0.28 in the LLAMA2-70B model. Note that using the above matrices does not change the computation as we still use a fast Hadamard kernel for the down-projection and out-projection layers.

## A.6 Round-to-Nearest Weight Quantization: Detailed Results

Table 9 shows the detailed results of QuaRot with GPTQ and round-to-nearest (RTN) weight quantization for both 6 and 8 bits on various tasks for LLAMA-2 models.

## A.7 FP16 Hadamard Transformation Ablation

We use FP32 online Hadamard transformation across all our experiments. Table 10 shows the results of using FP16 Hadamard transformation during the inference (for *down-projection* and *out-projection* layers). On LLAMA2-7B model, the results show <0.1 perplexity change on WikiText-2 and <0.6% averaged accuracy change on the zero-shot tasks, which we consider as noise. On LLAMA2-13B model, different Hadamard precisions have the same perplexities with 0.07% difference in the averaged zero-shot results. We conclude that the model will not be changed using different Hadamard precision.

Table 9: WikiText-2 Perplexity and zero-shot accuracy of QuaRot on the LLAMA-2 family using 4, 6 and 8-bits with GPTQ and RTN weight quantization and RTN activation quantization. For zero-shot tasks, we use PIQA (PQ), WinoGrande (WG), HellaSwag (HS), Arc-Easy (A-e), Arc-Challenge (A-c), and LAMBADA (LA).The Precision column shows the bitwidth for all inputs, weights, and KV-caches.

| Model | Method | Precision | PPL ↓ | PQ ↑ | WG ↑ | HS ↑ | A-e ↑ | A-c ↑ | LA ↑ | Avg. ↑ |
|-------|--------|-----------|-------|------|------|------|-------|-------|------|--------|
| | Baseline | FP16 | 5.47 | 79.11 | 69.06 | 75.99 | 74.58 | 46.25 | 73.90 | 69.82 |
| | QuaRot-RTN | INT4 | 8.37 | 72.09 | 60.69 | 65.40 | 58.88 | 35.24 | 57.27 | 58.26 |
| | | INT6 | 5.56 | 78.73 | 67.80 | 75.92 | 74.16 | 46.08 | 73.86 | 69.42 |
| 7B | | INT8 | 5.50 | 78.94 | 68.67 | 75.80 | 74.79 | 45.39 | 74.33 | 69.65 |
| | QuaRot-GPTQ | INT4 | 6.10 | 76.77 | 63.77 | 72.16 | 69.87 | 40.87 | 70.39 | 65.64 |
| | | INT6 | 5.52 | 78.45 | 69.46 | 75.60 | 74.45 | 46.50 | 74.19 | 69.77 |
| | | INT8 | 5.50 | 78.94 | 68.90 | 75.79 | 74.66 | 46.16 | 74.44 | 69.81 |
| | Baseline | FP16 | 4.88 | 80.47 | 72.22 | 79.39 | 77.48 | 49.23 | 76.75 | 72.59 |
| | QuaRot-RTN | INT4 | 6.09 | 77.37 | 67.32 | 73.11 | 70.83 | 43.69 | 70.66 | 67.16 |
| | | INT6 | 4.95 | 79.65 | 72.22 | 79.10 | 77.27 | 50.34 | 76.75 | 72.56 |
| 13B | | INT8 | 4.90 | 80.52 | 71.59 | 79.38 | 77.31 | 49.32 | 76.63 | 72.46 |
| | QuaRot-GPTQ | INT4 | 5.40 | 78.89 | 70.24 | 76.37 | 72.98 | 46.59 | 73.67 | 69.79 |
| | | INT6 | 4.92 | 79.98 | 72.69 | 79.17 | 77.78 | 49.74 | 76.27 | 72.60 |
| | | INT8 | 4.90 | 80.36 | 71.98 | 79.38 | 77.31 | 49.15 | 76.79 | 72.49 |
| | Baseline | FP16 | 3.32 | 82.70 | 77.98 | 83.84 | 80.98 | 57.34 | 79.58 | 77.07 |
| | QuaRot-RTN | INT4 | 4.14 | 80.69 | 75.14 | 79.63 | 77.57 | 51.71 | 77.02 | 73.63 |
| | | INT6 | 3.36 | 83.24 | 77.90 | 83.47 | 80.93 | 58.28 | 79.41 | 77.21 |
| 70B | | INT8 | 3.33 | 82.97 | 77.98 | 83.67 | 80.77 | 58.11 | 79.53 | 77.17 |
| | QuaRot-GPTQ | INT4 | 3.79 | 82.43 | 76.24 | 81.82 | 80.43 | 56.23 | 78.73 | 75.98 |
| | | INT6 | 3.35 | 82.13 | 77.66 | 83.63 | 80.89 | 57.08 | 79.70 | 77.02 |
| | | INT8 | 3.33 | 83.13 | 78.06 | 83.72 | 80.85 | 58.19 | 79.72 | 77.28 |

Table 10: Ablation on the precision of online Hadamard transformations for QuaRot. We use WikiText-2 perplexity as well as zero-shot tasks, explained in Section 5.3.

| Model | Method | Hadamard Precision | PPL ↓ | PQ ↑ | WG ↑ | HS ↑ | A-e ↑ | A-c ↑ | LA ↑ | Avg. ↑ |
|-------|--------|--------------------|-------|------|------|------|-------|-------|------|--------|
| | Baseline | - | 5.47 | 79.11 | 69.06 | 75.99 | 74.58 | 46.25 | 73.90 | 69.82 |
| 7B | QuaRot | FP32 | 6.10 | 76.77 | 63.77 | 72.16 | 69.87 | 40.87 | 70.39 | 65.64 |
| | | FP16 | 6.08 | 76.99 | 66.46 | 72.59 | 69.07 | 41.21 | 70.59 | 66.21 |
| | Baseline | - | 4.88 | 80.47 | 72.22 | 79.39 | 77.48 | 49.23 | 76.75 | 72.59 |
| 13B | QuaRot | FP32 | 5.40 | 78.89 | 70.24 | 76.37 | 72.98 | 46.59 | 73.67 | 69.79 |
| | | FP16 | 5.40 | 77.69 | 70.09 | 75.75 | 73.95 | 47.61 | 73.22 | 69.72 |

## A.8    LLAMA-3 Results

In this section, we show the accuracy of applying QuaRot for quantizing the LLAMA3-8B and LLAMA3-70B models. Table 11 shows the WikiText-2 perplexity of quantizing the LLAMA-3 models with QuaRot using 4-bit quantization. Compared to Table 1, we conclude that LLAMA-3 is more sensitive to quantization as we can see a higher gap between the quantized and FP16 models. Table 12 shows the accuracy results of those models on zero-shot tasks.

Table 11: WikiText-2 perplexity results on 4-bit quantization of LLAMA-3 models with 2048 sequence length. 128G shows the group-wise quantization with group size 128.

| Method | Weight Quantization | #Outlier Features | LLAMA-3 8B | 70B |
|---|---|---|---|---|
| Baseline | - | - | 6.14 | 2.86 |
| QuaRot | GPTQ | 0 | 8.16 | 6.66 |
| QuaRot-128G | GPTQ-128G | 0 | 7.36 | 5.51 |

Table 12: Zero-shot accuracy of LLAMA-3 models with 4-bit QuaRot on PIQA (PQ), WinoGrande (WG), HellaSwag (HS), Arc-Easy (A-e), Arc-Challenge (A-c), and LAMBADA (LA).

| Model | Method | PQ | WG | HS | A-e | A-c | LA | Avg. |
|---|---|---|---|---|---|---|---|---|
| LLAMA3-8B | FP16 | 80.74 | 72.77 | 79.06 | 77.82 | 53.33 | 75.63 | 73.22 |
| | QuaRot | 75.14 | 65.82 | 72.94 | 68.01 | 43.34 | 65.81 | 65.18 |
| LLAMA3-70B | FP16 | 84.66 | 80.51 | 84.89 | 85.86 | 64.25 | 79.47 | 79.94 |
| | QuaRot | 78.07 | 69.30 | 77.33 | 73.44 | 47.53 | 69.57 | 69.21 |

## A.9 Phi-3-mini-4k-instruct Results

In this section, we show the accuracy of applying QuaRot for quantizing the Phi-3-mini-4k-instruct model [Abdin et al., 2024]. Table 13 shows the accuracy results of the model in terms of perplexity and on zero-shot tasks.

Table 13: WikiText-2 Perplexity and zero-shot accuracy of QuaRot on the Phi-3-mini-4k-instruct model (revision = ff07dc01) using 4, 6 and 8-bits with GPTQ and RTN weight quantization and RTN activation quantization. For zero-shot tasks, we use PIQA (PQ), WinoGrande (WG), HellaSwag (HS), Arc-Easy (A-e), Arc-Challenge (A-c), and LAMBADA (LA).

| Model | Method | Precision | PPL ↓ | PQ ↑ | WG ↑ | HS ↑ | A-e ↑ | A-c ↑ | LA ↑ | Avg. ↑ |
|---|---|---|---|---|---|---|---|---|---|---|
| | Baseline | FP16 | 6.35 | 80.47 | 73.72 | 78.45 | 80.13 | 57.51 | 68.37 | 73.11 |
| | | INT4 | 11.69 | 68.39 | 58.64 | 60.60 | 65.87 | 39.25 | 43.99 | 56.12 |
| | QuaRot-RTN | INT6 | 6.78 | 79.54 | 73.01 | 77.46 | 79.21 | 55.12 | 67.53 | 71.98 |
| Phi-3-mini | | INT8 | 6.58 | 79.71 | 74.11 | 78.63 | 80.47 | 56.66 | 68.56 | 73.02 |
| | | INT4 | 7.85 | 75.35 | 67.88 | 72.95 | 72.98 | 48.12 | 60.78 | 66.34 |
| | QuaRot-GPTQ | INT6 | 6.63 | 79.54 | 72.69 | 78.50 | 79.42 | 56.74 | 68.85 | 72.67 |
| | | INT8 | 6.58 | 80.25 | 74.19 | 78.54 | 80.35 | 57.08 | 68.64 | 73.18 |

## A.10 Performance Analysis

We implement the attention mechanism using three routines: 1) **Init**: During the prefill stage, this routine initializes the cache from all the key and value vectors in the prefill. The attention output during prefill is computed directly using Flash Attention [Dao et al., 2022] since we already have access to dequantized keys and values. 2) **Append**: During decoding, this routine is called first to quantize the current keys and values and append them to the cache. 3) **Decode**: Finally, this routine is called during decoding with the current query vector. The routine computes the attention output using a quantized implementation of flash attention which can load the quantized cache and compute the final value vector.

**4-bit Linear and Attention Layers.** We benchmark our 4-bit linear layer which involves 4-bit matrix multiplication. For a given input of FP16, the layer optionally computes the Hadamard operation, then calls the quantization kernel to quantize and save the input in a sub-byte format. In the next step, the quantized weights and input are passed to the CUTLASS 4-bit GEMM kernel. Finally, the output is dequantized and cast back to FP16. Figure 7 shows the speedup of our 4-bit layer for different layer sizes where the layer sizes match the FFN linear layer sizes in LLAMA-2 models.

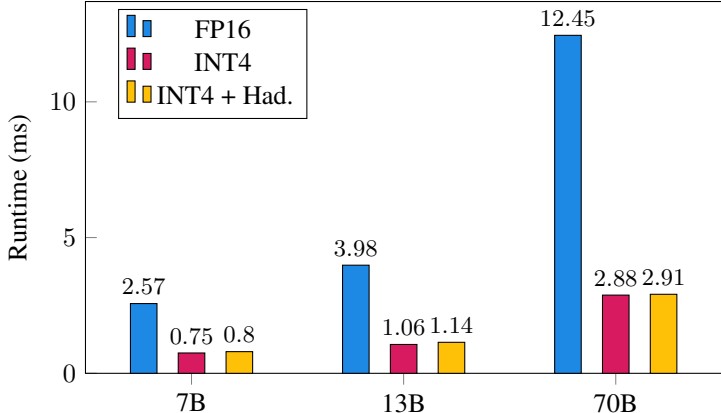

Figure 7: Performance of 16-bit and 4-bit linear layer for 2048 sequence lengths with and without online Hadamard transformation on a NVIDIA RTX 3090 GPU, averaged over 1000 runs. The matrix sizes correspond to the linear layer sizes in LLAMA-2 FFN blocks (i.e. $\mathbf{W}_{\text{down}}$). Here the batch size is 1, but the performance ratio holds for larger batches (see Table 14).

Our 4-bit linear layer gets 3.2x speedup relative to FP16 in the LLAMA2-7B model, and 4.3x on the LLAMA2-70B model. These numbers are for a batch size of 1, we find that scaling is approximately linear with batch size: more results in Table 14. We include the runtime with and without Hadamard operations, as $\mathbf{W}_{\text{up}}$ and $\mathbf{W}_{\text{gate}}$ do not require Hadamard transforms, whilst $\mathbf{W}_{\text{down}}$ does. We see that the Hadamard transform adds very little overhead to the forward pass at most 7% overhead.

We also compare the speed of performing append and decode routines for a single token given a cache of size 2047. This is equivalent to the cost of decoding the 2048-th token in a sequence. The comparison between the speed of FP16 and INT4 for different batch sizes and layer sizes is reported in Table 15. For the layer size used in LLAMA2-7B , our 4-bit implementation gets up to 1.72x improvement in speed for the larger batch sizes (e.g. from 16 onwards). The 4-bit cache is slower than FP16 for smaller batch sizes (e.g. up to 8). Note that this is intuitive as the main benefit of the 4-bit cache is reducing the I/O cost. A speed up is only visible if this reduction is more significant than the quantization overhead which happens for either larger batch sizes or longer sequences.

Table 14 shows the results of benchmarking our 4-bit linear layer. The layer sizes are extracted based on the linear layer sizes in LLAMA-2 models (for out-projection and down-projections). We apply both FP16 and FP32 Hadamard transformations and show the runtime on NVIDIA RTX GPU using 2048 sequence lengths. Table 15 shows the results of decoding a single token in the attention layer when we apply KV-cache quantization. We extract the size of the attention layer based on the LLAMA-2 models.

| Layer Size | Batch Size | FP16 | INT4 | INT4 + FP32 Had | INT4 + FP16 Had |
|---|---|---|---|---|---|
| 4096x4096 | 1 | 1.043 | 0.370 | 0.409 | 0.403 |
|  | 2 | 1.902 | 0.696 | 0.790 | 0.789 |
|  | 4 | 3.715 | 1.361 | 1.522 | 1.529 |
|  | 8 | 7.200 | 2.675 | 2.999 | 3.011 |
|  | 16 | 14.508 | 5.357 | 5.973 | 5.976 |
|  | 32 | 29.029 | 10.641 | 11.900 | 11.911 |
| 5120x5120 | 1 | 1.418 | 0.464 | 0.552 | 0.547 |
|  | 2 | 2.918 | 0.937 | 1.100 | 1.097 |
|  | 4 | 5.852 | 1.888 | 2.206 | 2.207 |
|  | 8 | 11.465 | 3.809 | 4.428 | 4.422 |
|  | 16 | 22.807 | 7.547 | 8.755 | 8.759 |
|  | 32 | 45.312 | 15.019 | 17.417 | 17.440 |
| 8192x8192 | 1 | 3.696 | 0.997 | 1.084 | 1.083 |
|  | 2 | 7.191 | 1.944 | 2.099 | 2.099 |
|  | 4 | 14.236 | 3.918 | 4.208 | 4.207 |
|  | 8 | 28.508 | 7.944 | 8.460 | 8.415 |
|  | 16 | 57.814 | 15.793 | 16.859 | 16.871 |
|  | 32 | 115.462 | 31.693 | 33.780 | 33.791 |
| 11008x4096 | 1 | 2.569 | 0.749 | 0.798 | 0.801 |
|  | 2 | 5.027 | 1.478 | 1.555 | 1.558 |
|  | 4 | 9.752 | 2.990 | 3.140 | 3.144 |
|  | 8 | 19.696 | 6.031 | 6.296 | 6.306 |
|  | 16 | 38.883 | 11.978 | 12.503 | 12.527 |
|  | 32 | 78.320 | 23.874 | 24.935 | 24.974 |
| 13824x5120 | 1 | 3.983 | 1.063 | 1.142 | 1.139 |
|  | 2 | 7.869 | 2.148 | 2.291 | 2.293 |
|  | 4 | 15.410 | 4.340 | 4.616 | 4.614 |
|  | 8 | 30.761 | 8.719 | 9.231 | 9.240 |
|  | 16 | 61.203 | 17.318 | 18.345 | 18.343 |
|  | 32 | 122.926 | 34.816 | 36.953 | 36.940 |
| 28672x8192 | 1 | 12.450 | 2.881 | 2.911 | 2.911 |
|  | 2 | 25.391 | 5.828 | 5.892 | 5.896 |
|  | 4 | 50.742 | 11.938 | 11.947 | 11.976 |
|  | 8 | 101.290 | 24.186 | 24.202 | 24.216 |
|  | 16 | 202.909 | 48.238 | 48.325 | 48.356 |
|  | 32 | 406.344 | 96.761 | 97.044 | 96.892 |

Table 14: Performance of 4-bit linear layer for 2048 sequence lengths with and without online Hadamard transformation on a NVIDIA RTX 3090 GPU. The matrix sizes correspond to the linear layer sizes in LLAMA-2 models. We averaged over 100 runs and report the numbers in milliseconds.

| head_num x head_dim | Batch Size | FP16 | INT4 | INT4 + FP32 Had | INT4 + FP16 Had |
|---|---|---|---|---|---|
| 32x128 | 1 | 0.713 | 1.033 | 1.163 | 1.117 |
| | 2 | 0.723 | 1.035 | 1.168 | 1.122 |
| | 4 | 0.781 | 1.033 | 1.168 | 1.118 |
| | 8 | 0.984 | 1.042 | 1.173 | 1.126 |
| | 16 | 1.348 | 1.018 | 1.153 | 1.102 |
| | 32 | 2.098 | 1.168 | 1.247 | 1.216 |
| 40x128 | 1 | 0.712 | 1.026 | 1.157 | 1.106 |
| | 2 | 0.726 | 1.035 | 1.173 | 1.121 |
| | 4 | 0.831 | 1.038 | 1.166 | 1.115 |
| | 8 | 1.065 | 1.048 | 1.181 | 1.128 |
| | 16 | 1.525 | 1.021 | 1.153 | 1.102 |
| | 32 | 2.480 | 1.244 | 1.320 | 1.287 |
| 64x128 | 1 | 0.715 | 1.028 | 1.160 | 1.108 |
| | 2 | 0.780 | 1.034 | 1.171 | 1.117 |
| | 4 | 0.984 | 1.034 | 1.171 | 1.120 |
| | 8 | 1.361 | 1.048 | 1.182 | 1.130 |
| | 16 | 2.071 | 1.147 | 1.223 | 1.192 |
| | 32 | 3.563 | 1.566 | 1.645 | 1.612 |

Table 15: Performance of decoding a single token with 4-bit KV cache for the attention layer for 2048 sequence lengths with and without online Hadamard transformation on an NVIDIA RTX 3090 GPU. We evaluate generating the last token when the 2047 tokens are already cached in the attention. We extract the number of heads (head_num) and their dimensions (head_dim) based on different LLAMA-2 models. We averaged over 100 runs to report the numbers in milliseconds.

Tables 16 and 17 show the detailed speedups and memory saving of a single transformer block for QuaRot on LLAMA2-7B model using NVIDIA RTX 3090 GPU.

| Model | Batch Size | Speedup |
|---|---|---|
| LLAMA2-7B | 1 | 1.97× |
| | 4 | 2.06× |
| | 16 | 2.11× |
| | 32 | 2.14× |
| | 64 | 2.16× |
| LLAMA2-70B | 1 | 3.16× |
| | 4 | 3.27× |
| | 16 | 3.32× |
| | 32 | 3.33× |

Table 16: Time-to-first-token (prefill) speedup of each transformation block of LLAMA-2 models in QuaRot (over the FP16 model) on NVIDIA RTX 3090 GPU. We use 2048 sequence lengths with different batch sizes.

| Model | Batch Size | Sequence Length | Baseline (GB) | QuaRot (GB) | Saving Factor |
|---|---|---|---|---|---|
| LLAMA2-7B | 1 | 256 | 0.392GB | 0.108GB | 3.63× |
| | | 512 | 0.396GB | 0.108GB | 3.66× |
| | | 1024 | 0.404GB | 0.110GB | 3.66× |
| | | 2048 | 0.419GB | 0.114GB | 3.67× |
| | | 4096 | 0.451GB | 0.125GB | 3.60× |
| | 16 | 256 | 0.464GB | 0.128GB | 3.63× |
| | | 512 | 0.528GB | 0.144GB | 3.66× |
| | | 1024 | 0.655GB | 0.177GB | 3.70× |
| | | 2048 | 0.908GB | 0.244GB | 3.72× |
| | | 4096 | 1.416GB | 0.378GB | 3.75× |
| LLAMA2-70B | 1 | 256 | 1.605GB | 0.409GB | 3.92× |
| | | 512 | 1.606GB | 0.409GB | 3.92× |
| | | 1024 | 1.608GB | 0.410GB | 3.92× |
| | | 2048 | 1.612GB | 0.411GB | 3.92× |
| | | 4096 | 1.620GB | 0.413GB | 3.92× |
| | 16 | 256 | 1.626GB | 0.418GB | 3.89× |
| | | 512 | 1.642GB | 0.422GB | 3.89× |
| | | 1024 | 1.674GB | 0.430GB | 3.89× |
| | | 2048 | 1.738GB | 0.447GB | 3.89× |
| | | 4096 | 1.865GB | 0.480GB | 3.89× |

Table 17: Peak Memory usage (in GB) for decoding a single token on a single transformation block of LLAMA-2 models with KV caches of different lengths and with different batch size.

