# OpenReview forum: "QuaRot: Outlier-Free 4-Bit Inference in Rotated LLMs"
_NeurIPS.cc/2024/Conference — NeurIPS 2024 poster_

### Official Review · Reviewer_tVLH · 2024-06-15

**Soundness:** 4
**Presentation:** 4
**Contribution:** 4
**Rating:** 8
**Confidence:** 5

**Summary:**

The paper focusses on improving post-training-quantization for LLMs. Specifically, they combine insights into incoherence processing from the two #QUIP papers and rotational invariance from SliceGPT to improve accuracy after quantization. The insight is that LLMs are, or can be made, invariant to rotations on the residual, which can then be exploited by multiplying the residual with random rotation matrices (and it's inverse), which approximately reduce outliers in the activations and weights. A part of this method comes at the cost of no extra overhead, as the weight matrices are absorbed into the layers. The paper also introduces a specific form of rotation matrix, the Hadamard matrix (which was also used in QUIP# in several other parts of the network. These Hadamard matrices cannot be absorbed, but can be efficiently executed with the Walsh-Hadamard matrix multiplication algorithm, incurring only little overhead.
The paper does extensive experiments on the LLamaV2 model showing the quantization is significantly improved after the rotation operations. The authors also combine the method with other methods that improve performance, such as GPTQ and grouped-quantization, showing that the method can be combined with other quantization-improvement methods.
The authors also show actual speed-ups from their 4bit kernels.

**Strengths:**

The idea of combining SliceGPT and QUIP was out there hanging in the ether, as the two papers compliment each other very neatly. I might expect more authors in this field to do similar work. Regardless, I believe this work is novel at this point.
The impact of this work is potentially quite large, as it allows LLMs to significantly improve in speed by having lower-bit-width quantization much more available. On top of this, the algorithms are pretty easy to apply. The paper reads as a full state-of-the-art quantization pipeline that can be applied on LLMs in the PTQ setting. I can see this being applied in many practical LLM settings.
The paper is well-written and easy to follow. It does not claim anything it doesn't do, and is factual without scruples.
The experiment section is solid, and the ablations in the appendices good. The code has been released, and I have been able to verify some of the numbers from the authors. The authors went the extra mile showing not only their theoretical gains, but also speed-ups in practice.

**Weaknesses:**

- Emoji's in titles should get desk-rejected ;)
- Table 2 does not really compare to anything. There's plenty of papers that do 4 bits, you have the original GPTQ paper, LLM-QAT, ATOM, OmniQuant, SmoothQuant, simple RTN. it's pretty tough to contextualize the results here; but zero-shot is quite important for a proper evaluation

**Questions:**

Is it possible to put more comparisons with other methods into the tables?

In Table 7, you show that the GPTQ algorithm for 7B gives worse performance than RTN. I also validated that this happens in your codebase. How is it possible that GPTQ gives a worse performance than RTN? Theoretically, it should reduce the MSE for each layer, and never really give a worse performance. Is your implementation correct?

Editorial notes:
- 140 consider linking computational invariance to the relevant section

**Limitations:**

They have adequately addressed the limitations.

---

> ### Author Rebuttal · Authors · 2024-08-05
>
> Thanks for your comments.
>
> > Emoji's in titles should get desk-rejected ;)
>
> The emoji is there just because we wanted to give a hint on how QuaRot should be pronounced :-)
>
> > Table 2 does not really compare to anything. Is it possible to put more comparisons with other methods into the tables?
>
> We compared the WikiText results with other papers in Table 1. However, zero-shot experiments are presented to show that our scheme works well over various tasks. We will evaluate other schemes on the same set of tasks and include them in the next version.
>
> > In Table 7, you show that the GPTQ algorithm for 7B gives worse performance than RTN.
>
> We agree that this is surprising. This may be because of using GPTQ with default parameters and not tuning the parameters in the algorithm (some of them are mentioned here: https://github.com/IST-DASLab/gptq?tab=readme-ov-file#new-features).

---

> > ### Comment · Reviewer_tVLH · 2024-08-12
> > **Final review**
> >
> > I have read all reviewer comments and rebuttals, and could not find anything that would change my mind. Because of this, I'm keeping my score of strong accept. I think this method is very worthwhile, and likely going to be much cited and used in the field of LLM quantization. I believe it is significantly novel, and not just a trivial combination of two methods.

---

> > > ### Author Response · Authors · 2024-08-13
> > > **Official Comment by Authors**
> > >
> > > Thank you so much for your positive comments. We appreciate your  time and effort on our paper.

---

### Official Review · Reviewer_hgDj · 2024-07-04

**Soundness:** 3
**Presentation:** 3
**Contribution:** 2
**Rating:** 5
**Confidence:** 3

**Summary:**

This paper applies the random Hadamard transform (RHT) in strategic places in the GPT architecture to improve the quality of weight and activation quantized models. The RHT is a fully invertible (up to machine epsilon) transformation that effectively concentrates matrices. The authors claim that applying the RHT to activations reduces the effect activation outliers have on activation quantization performance. The authors show that their method can be used for INT4 weight and activation quantization without significant degradation, which enables higher throughput inference in compute-bound scenarios.

**Strengths:**

The empirical results appear to be pretty good. Quantizing both the weights and activations to INT4 using QuaRot results in minimal degradation and enables up to a >3x speedup during prefill.

**Weaknesses:**

My main concern with this paper is that it combines methods from existing works (RHT from QuIP#, GPTQ from GPTQ, "Comptuational Invariance" from SliceGPT) instead of introducing something actually novel and new. Those works independently showed that their respective ideas worked, so it is not surprising that combining them also works. Fusing the Hadamard transformation into the weight matrix and rotating the activations is also not something "new", as this is actually how QuIP# does inference (it is cheaper to rotate the activations than the weights in small-batch settings).

Regarding the actual method, QuaRot essentially does GPTQ + Incoherence Processing on the weight matrix, which QuIP showed was optimal within a certain class of rounding methods. However, to the best of my knowledge, there is no optimality result for doing Incoherence Processing + RTN, which is what QuaRot does on the activations. In fact, the distortion for quantizing a Gaussian (the result of the RHT) with a uniform scalar quantizer (INT4) is actually pretty bad relative to the distortion for quantizing other common distributions with a uniform scalar quantizer.

**Questions:**

1. What is the comptuational cost of each of the steps in QuaRot? The matrix multiplication is the dominating factor, but the Hadamard transformation is not completely free. If you used the RHT as presented in QuIP#, the RHT can actually be quite slow for "bad" matrix dimensions where the non-power-of-2 component is quite large (eg 172 in Llama 2 7B). How would you solve this in practice?
2. The RHT requires "mixing" activations across the input dimension, making it costly to use with tensor parallelism since you need to send lots of data across a potentially slow bus. Have you run any experiments on the effectiveness of QuaRot when the RHT applied "per slice" (eg 1/8 of the activation dimension for 8 GPUs)? How well does the RHT work here?
3. RE the uniform scalar quantizer comment above, FP4 would probably work better for quantizing a Gaussian. Can you run an experiment with FP4 activations and weights and see if the quality improves?

**Limitations:**

Yes

---

> ### Author Rebuttal · Authors · 2024-08-05
>
> Thank you so much for your comments.
>
> > this paper is that it combines methods from existing works
>
> Here, we explain our contributions and the main differences between QuaRot and the existing work.
>
> 1. SliceGPT: SliceGPT focuses on compressing LLMs by slicing the weights. Although we use the same principle (computational invariance) from SliceGPT, we apply RHT as an orthogonal transformation which is different from what they use  (Principal components) and especially suited to quantization because it makes the input distribution more uniform when it has some outliers (Fig. 1). In addition, SliceGPT needs to undo the transformations in the shortcuts as they use PCA over the input of each linear layer and QuaRot uses a single RHT for the whole model which eliminates the need for shortcut transformations.
>
> 2. GPTQ and QuIP#: GPTQ and QuIP# focus on weight-only quantization while we focus on quantizing weights, activations, and KV-caches.  Quantizing weights only has the practical impact of having a smaller footprint for the model. Quantizing activations additionally saves working space at inference time. Quantizing KV-cache has a huge practical advantage of much higher token throughput (especially in long-context generation). Although we also use RHT (similar to QuIP#), we remove the overhead of applying such transformations (and their inverse) by fusing the RHT into the adjacent layers and applying it after RoPE for KV-cache quantization.
>
> 3. Practical usage: We developed a set of CUDA kernels for our work and got up to 3.33x speedups and 3.89x memory saving over the FP16 version. Compared to the previous 4-bit quantization work [1, 2], we have simpler kernels as we do not need complex quantization schemes after applying RHT.
>
> >as this is actually how QuIP# does inference
>
> There are a number of important differences between QuIP# and QuaRot which we summarize here:
> QuIP# applies the Haramard transformations before quantization for each weight matrix in the model. During the inference, the inverse of such transformations should be applied to undo the transformation which adds overhead in each linear module. However, QuaRot does not need such extra transformations as we fuse the Hadamards into the previous layer (Stage 1a. of the Method section). This enables us to 1) apply fewer Hadamard transformations (only 1 in the MLP and only a single intra-head in the attention) during the inference and 2) remove the input outliers so we can quantize both weights and inputs to 4-bits.
>
>
> >What is the comptuational cost of each of the steps in QuaRot?
>
> This is one of the advantages of QuaRot over existing work. In the first step, QuaRot fuses the layer-norm and replaces them with RMSNorm (which preserves any orthogonal transformations on the input). This helps us to completely remove the overhead of applying RHT by fusing it into the previous linear weights. This guarantees that the input of each linear layer is already on the (randomized) Hadamard space. We undo the input Hadamard by fusing its inverse to the weight of the linear layer so we will have XQQW = XW. In total, for every decoder block, we need to apply a single Hadamard transformation in the MLP (after non-linearity) and only a single intra-head in the attention (before Our-proj). For such transformations, we use a fast Hadamard kernel which leads to minimal overhead (Table 15).
>
> >Have you run any experiments on the effectiveness of QuaRot when the RHT applied "per slice"?
>
> We did not apply the tensor-parallel as we focus on LLM inference where TP is a less common setting in that case. However, consider an input matrix of size `BxH` where B is the `batch-size*sequence_len` and `H` is the hidden dimension of the model. If the matrix is split across the rows which is suitable for RowMajor matrices (for example, we have 8 matrices of size `B/8xH`), the Hadamard of size `H` could be applied on each chunk independently. In the case of splitting the columns, one can apply smaller Hadamard matrices on each submatrix of `BxH/8` independently and quantize each block separately.
>
> >FP4 would probably work better for quantizing a…
>
> We ran the following experiment for LLaMa2 models with W4A4KV4 just now, comparing INT4 vs MXFP4 with group-size 128. The MXFP4 format [3] is an alternative to the standard FP4 format which shares an additional common scale within the group, so should perform slightly better than FP4 at a cost of an extra 1/16th of a bit per weight (see [3], Section 5.1 for details). The results show that MXFP4 performs worse than INT4 across the LlaMa2 model sizes. This perhaps surprising result implies the post RHT distributions are more uniform than purely Gaussian.
>
>
> | Model |  INT4 | MXFP4|
> |----|------|------|
> |7B| 5.93|6.52|
> |13B| 5.26 |5.71|
> |70B| 3.61 |3.82|
>
> **References**:
>
> [1] https://arxiv.org/abs/2310.19102
>
> [2] https://arxiv.org/abs/2310.09259
>
> [3] https://www.opencompute.org/documents/ocp-microscaling-formats-mx-v1-0-spec-final-pdf

---

> > ### Comment · Reviewer_hgDj · 2024-08-09
> > **Comments**
> >
> > My point regarding SliceGPT, GPTQ, and QuIP# is that you're basically taking the RHT from QuIP# and fusing it into an adjacent layer where you can, and then quantizing the activations with nearest rounding. This is a nice implementation detail that improves throughput in compute bound scenarios. However, the only contribution here is fusing the Hadamard transform, which just doesn't seem to be that novel (this essentially amounts to saying linear operations can be fused). Nearest rounding is also very simple, and the results you posted above with MXFP4 and INT4 suggest that nearest rounding is suboptimal for the activations. Post-RHT distributions are very close to Gaussian (you can verify this with a QQ plot), so MXFP4 should result in lower distortion for the activations, but perplexity is higher. I would have liked to see some analysis on what the best way to round the activations is. Are there any rounding algorithms that can improve over direct rounding while still being fast enough to not have large overhead during inference?
> >
> > Regarding tensor parallelism, running the RHT per batch of data would be emulating data parallelism, while running the RHT on slices of input channels would be tensor parallelism. The reason why I asked is because the main benefit of QuaRot seems to be in compute bound scenarios where we can benefit from INT4 hardware, such as large batch inference. In memory bound scenarios where the activations are small, the memory savings from quantizing activations aren't that large. However, large batch inference usually means doing some sort of sharding like DP, TP or even both. Using TP with the RHT would mean having to use a smaller Hadamard matrix, which would reduce quality. The incoherence bounds from applying the RHT depend on the matrix dimension. Can you run an experiment where you quantize the weights as 8 separate slices along the channel dimension? I would like to see how much degradation there is by doing this. If the matrix dimension is large enough, this might not actually make a difference.
> >
> > I may have overlooked the contribution on using the RHT to help quantize the KV cache, which I think is new and hasn't been done before. For that reason, I will raise my score to a 5. The paper is still more of an implementation optimization paper on existing methods, so outside of the KV cache part the contribution is limited.

---

### Official Review · Reviewer_QCed · 2024-07-04

**Soundness:** 4
**Presentation:** 3
**Contribution:** 3
**Rating:** 7
**Confidence:** 4

**Summary:**

The authors provide a framework for W4A4KV4 quantization of LLMs, leading to computing and peak memory improvement while maintaining good performance on language generation and zero-shot tasks. They achieve this by introducing randomised Hadamard transformations at both weight and activations of transformer blocks. They effectively build on the ideas of incoherence processing and computational invariance that have shown that applying orthogonal transformation to weight matrices in LLMs allows them to be quantized with smaller errors while ensuring the network output is unchanged.

**Strengths:**

* Achieving good accuracy in W4A4KV4 LLM quantization is a challenging problem. It is commendable that the authors achieve good accuracy in zero-shot and language generation tasks in such an aggressive quantization regime with significant compute and peak memory savings.
* The paper is very well written, with a concise and clear background section and readable graphs (although some colour-coding of the boxes would be helpful).
* Applying the online Hadamard transformation to all activations is an interesting idea that improves the quantizabilty of activation and KV cache. Given the complexity of rotational embeddings, the authors have also taken great care in consistently applying these transformations.
* The authors have conducted very good and extended quantization ablation studies. In the current landscape of numerous quantization hacks and choices for LLMs, this research is crucial in understanding the effect of each choice.

**Weaknesses:**

* The motivation for using Hadamard matrices as orthogonal matrices is not well-established. Given that incoherence processing and computation invariance have established that orthogonal matrices can be used to improve quantization/pruning, I would like to see a more coherent and concise motivation for using Hadamard matrices. I can tell from section 3.1 that they are computationally efficient, but this is mentioned as a side note rather than the reason for using them. Building on this, it is still unclear to me from sections 3 & 4 if Hadamard matrices are the authors' contribution or taken from previous LLM quantization work.
* The comparison to other relevant work lacks details. In Table 1, the authors should take better care to ensure that the quantization granularity of weights, activations, and cache is coherent between their method and the other benchmarks. For example, for a fair comparison, they should use SmoothQuant-O1 (see Table 2 of the paper). Even if the authors use the correct quantization scheme, they should detail any differences between their method and the competing ones either in section 5.1 or the appendix. LLM quantization methods are becoming increasingly complex. Therefore, it is important that there is transparency when comparing perplexity to other methods.
* Why not include the same benchmark methods from Table 1 to Table 2?

**Questions:**

- Figure 3. What does the (α) mean in the box? Is it supposed to be a diagonal?
- Have you considered listing all the quantization bitwidths/granularities weights/activations, etc., mentioned in section 5 in a table? See Table 2 of SmootQuant.
- What operation does the Hadamard block do in Figs. 3&6. Based on my calculations, it is a post: $Y = X H^T$ . Please add more information about what these blocks do for each graph.
- Are all results in Table 1 for all methods with W4A4KV4 and the same granularity everywhere?
- Is the perplexity in Table 1 for the test or validation set?
- Line 280 explicitly mentions that this is about a single transformer block rather than saying it later in the paragraph. One needs to look in the appendix or the brackets below.
- Figure 4: Please mention that these are the averages of 100 runs as you do in the appendix. What is the STD of these runs?
- Table 9: Are weight and activation quantized to the same bitwidth? What about KV cache? Please be more explicit about the bit width choices per tensor in all tables and figures.
- Appendix A.6: why is this ablation study there? What is the take-aways?

**Limitations:**

The authors recognise the limitations of the method and provide a roadmap of improvements in the conclusion section.

---

> ### Author Rebuttal · Authors · 2024-08-05
>
> Thank you very much for your comments and encouragement. We agree that we can improve the readability of our results and include more details in our Tables. We have included these details in our manuscript.
>
> >Figure 3. What does the (α) mean in the box? Is it supposed to be a diagonal?
>
> Yes, this means the diagonal matrix.
>
> >What operation does the Hadamard block do in Figs. 3&6. Based on my calculatio…
>
> This is an online (non-fused) Hadamard, as described in Section 4.
>
> >Are all results in Table 1 for all methods with W4A4KV4 and the same granularity everywhere?
>
> Yes.
>
> >Is the perplexity in Table 1 for the test or validation set?
>
> The PPL in Table 1 is computed over the test  set.
>
> >Figure 4: Please mention that these are the averages of 100 runs as you do in the appendix. What is the STD of these runs?
>
> The std of the runs are <0.1 in all cases.
>
> >Table 9: Are weight and activation quantized to the same bitwidth? What about KV cache? Please be more explicit about the bit width choices per tensor in all tables and figures.
>
> Yes. all the weights, activations, and KV-caches are in the same precision in Table 9.
>
> >Appendix A.6: why is this ablation study there? What is the take-aways?
>
> The main take-away from this section is that in 8- and 6-bits RTN performs essentially as well as GPTQ when we use orthogonal transformations in QuaRot.

---

> > ### Comment · Reviewer_tVLH · 2024-08-08
> > **Rebuttal incomplete**
> >
> > Did you per-chance forget to post an answer to the 'weaknesses' indicated by the reviewer? o_O

---

> ### Author Response · Authors · 2024-08-08
> **Weakness answers**
>
> Thank you so much for your reply. Sorry that we didn’t post that part here. I have added them now:
>
> >The motivation for using Hadamard matrices as orthogonal matrices is not well-established.
>
> There are a few steps things we should add to justify using Hadamard matrices in our work:
>
> 1. Using computational Invariance, we can remove (almost all of the) overhead for our transformation if we use an orthogonal linear transformation. This forces us to use an orthogonal matrix as we can fuse them into the weights without needing to apply them during the inference.
>
> 2. By 1, we need to find a proper orthogonal matrix that makes the activations easy-to-quantize. We tried random orthogonal matrices which resulted in non-trivial, but also not great, accuracy on wikiText (see Section A.5 for such results).
>
> 3. After that, we have seen some related work that uses Hadamard transformations for quantizing the weights or during training (see Related work section - lines 60-65 for that). We found that using randomized Hadamard matrices with computational invariance results in a proper accuracy for almost free overhead.
>
> 4. Finally, for some linear layers (down-proj and out-proj), we cannot fuse orthogonal transformations. For that, we use exact Hadamard transformations. Fortunately, such transformations have fast CUDA kernels with low overhead (see Table 14 for that).
>
> > The comparison to other relevant work lacks details.
>
> We mentioned our experiment setup (with quantization details) in Section 5 (see lines 231-240). For the other work, we just used the existing numbers from related works (mentioned in Table 1) as some of the schemes (for example SmoothQuant) are not presented for 4-bit case. However, there is a marginal difference (>2.6 PPL points)  in all other cases when we quantize both weights and activations using 4-bit. For Atom, we carefully selected our details (for the group-wise case) to make sure we have a fair comparison against them (based on the numbers in their paper). We should note that Atom preserves some outlier features in higher precision where we do not do such outlier selections.
>
> >Why not include the same benchmark methods from Table 1 to Table 2?
>
> The main goal of this table is to show that QuaRot performs properly on Zero-Shot tasks.  In addition, we couldn’t find a proper selection of the tasks in all related work to include in our table for a fair comparison. Finally, as running Zero-Shot results are time-consuming for the models, we plan to implement all the related schemes and run the Zero-Shot experiments for them on the same quantization configuration for the next version of the paper.

---

> > ### Comment · Reviewer_QCed · 2024-08-12
> > **Comparison to other relevant work lacks details**
> >
> > I see now that you have taken the SmoothQuant and OmniQuant numbers from Table A23 of the OmniQuant paper. Whereas taking the OmniQuant numbers from their author makes sense, it would be better to re-implement SmoothQuant yourself and validate these numbers, given that the authors of SmoothQuant did not provide these.
> >
> > However, there is a significant difference in the implementation of multi-head attention between QuaRot and SmoothQuant. According to Fig.6 and section 2, the batched matmul is kept in FP16 in your method. Meanwhile, in SmoothQuant (see Fig. 6 in their paper), the BMM operations are kept in INT8. How do you account for this in your comparison? OmniQuant also quantizes the self-attention BMMs with the exception of the SoftMax output.
> >
> > Even if you have considered these differences already, a discussion and analysis of the quantization differences is imperative when comparing to other methods.

---

> ### Author Response · Authors · 2024-08-13
> **Comparison to other work**
>
> Thank you so much for your point.
>
> SmoothQuant is designed for INT-8 case and we saw a huge accuracy gap (>77.5 PPL point on 7B model) between it's 4-bit case and the FP16 model and this should not be only because of INT8 multi-head attention (as SmoothQuant shows that INT8 quantization of all layers + multi-head attention can preserves the accuracy).
>
> However, we agree with your point and we will run such experiments from our side for the next version.

---

> > ### Comment · Reviewer_QCed · 2024-08-13
> >
> > I agree with you that I do NOT think that SmoothQuant can close the accuracy gap, but it's good practice to be clear about quantization setup differences when comparing to other methods, not just for fairness but also for helping readers navigate the increasingly complex choices when quantizing self-attention.

---

### Official Review · Reviewer_rEuG · 2024-07-11

**Soundness:** 3
**Presentation:** 3
**Contribution:** 3
**Rating:** 7
**Confidence:** 4

**Summary:**

The paper introduces QuaRot, a novel quantization approach for Large Language Models (LLMs) that utilizes Hadamard transformations to address the challenge of outlier features in activations, weights, and KV caches. By incorporating these transformations, QuaRot enables the entire model, including activations and caches, to be quantized to 4 bits without significant loss in performance. This method improves both the computational efficiency and memory usage during model inference, preserving up to 99% of zero-shot performance in tests with the LLAMA2-70B model.

**Strengths:**

QuaRot addresses quantization of both weights, activations and KV caches, making the approach practical in the real-world scenarios.

The application of randomized Hadamard transformations helps in effectively managing outlier features, allowing for lower bit representation without performance degradation.

The method achieves substantial efficiency improvements, evidenced by up to 3.33× speedups in prefill operations and significant memory savings during the decoding stage.

**Weaknesses:**

Although the method sounds good in terms of the computational invariance, the experimental results are not impressive to some extent. Based on Table 2, there is a clear margin between the FP16 baseline and the proposed QuaRot on 7B, 13B and 70B settings.

The distinction between the proposed randomized Hadamard transformations and the Hadamard quantization method in Xi et al “Training Transformers with 4-bit Integers” should be elaborated.

**Questions:**

How to extend your method if the model adopts different normalization methods, like layer normalization or batch normalization. Also, why INT4 but not INT2 or INT1 as QuaRot can handle the latter two too?

Why the RMSNorm is not quantized? In the case of normalization layer quantization, how much the acceleration performance of the model and its performance on the datasets will be affected?

Furthermore, are the biases in the linear layers and the residual connections quantized to INT4?

Please specify your meaning of online Hadamard transform on line 38.

The consideration of the Hadamard transformations increases the difficulty of implementing the quantization kernel. Please explain more about how the quantization kernel is implemented based on the CUTLASS library to achieve real speedup?

In Fig. 3, (\alpha) should be diag(\alpha)? Also, line 122, the Q^T on output matrix becomes Q in Fig. 3?

**Limitations:**

The INT4 QuaRot still incurs significant drops from INT8 and FP16.

---

> ### Author Rebuttal · Authors · 2024-08-05
>
> Thank you very much for your comments!
>
> >The distinction between the proposed randomized Hadamard transformations and the Hadamard quantization method in Xi et al “Training Transformers with 4-bit Integers” should be elaborated.
>
> We acknowledged the above work in our paper (lines 63-64) and described the fact that they have used the Hadamard transformations during training. We have edited the manuscript to emphasize that they use “exact” Hadamard (not randomized Hadamard) transformation.
>
> >How to extend your method if the model adopts different normalization methods, like layer normalization or batch normalization.
>
> The main difference between LlamaRMSNorm (LRMSN) and LayerNorm (LN) is that the former does not have mean subtraction.
> However, as shown in [1] (equation 1), mean subtraction could be written as another linear transformation where we can fuse into the previous linear layer and replace LN with our RMSNorms. To our knowledge, Batch-Normalization is not used in any modern LLMs (which is the main focus of our work).
>
> >Also, why INT4 but not INT2 or INT1 as QuaRot can handle the latter two too?
>
> QuaRot could be applied with any precision (for example, we presented weight-only results for lower precision like INT3 in the submission -> Table 7). However, we focus on INT4 due to hardware support, which allows us to get direct speedups. We should note that Quantizing weights to fewer than 4 bits is possible (Table 7), but this does not always translate to a speedup due to lack of hardware support in all scenarios. In practice, it means using 4 bits but still sacrificing quality unless custom packing/unpacking support is implemented to reap the benefit.
>
> >Why the RMSNorm is not quantized?
>
> We keep the RMSNorm in high precision to avoid numerical instability (we need to calculate the l_2 norm of the input vectors and divide the input by it). However, keeping those modules in high precision does not meaningfully affect the end-to-end FLOP reduction as they are <0.4% of the total FLOPs in the baseline model [2].
>
> >Are the biases in the linear layers and the residual connections quantized to INT4?
>
> In LLaMa models, the linear layers do not have biases. However, biases could be fused into the matmul kernel to be applied just after dequantization. We also do not quantize the shortcuts as we focus on the computational bottlenecks of the model (which are linear layers).
>
> >Please specify your meaning of online Hadamard transform on line 38.
>
> Throughout the paper, we make the distinction between “fused” Hadamard transforms (i.e where the weight is modified as WH) and “not fused” ones, which we call “online” as they must be executed independently.
>
> >Please explain more about how the quantization kernel is implemented based on the CUTLASS library to achieve real speedup?
>
> Right now, we use separate kernels for this. We  Apply Hadamard -> Quantize -> MatMul -> DeQuantize. The quantization part is done on PyTorch.  However, the first two operations could be fused into a single kernel. We described these details in the Appendix A.10.
>
> >In Fig. 3, (\alpha) should be diag(\alpha)? Also, line 122, the Q^T on output matrix becomes Q in Fig. 3?
>
> Thanks for spotting that, we have amended the manuscript. For the Q^T, we apply exact Hadamard transformation from the left and call it by H (instead of Q) so the Q in the down-proj layer of Fig. 3 is the randomized Hadamard transform we apply to the output of that linear layer (to form YQ).
>
>
> **References**:
>
> [1] https://arxiv.org/pdf/2401.15024
>
> [2] https://huggingface.co/spaces/MrYXJ/calculate-model-flops

---

> > ### Comment · Reviewer_rEuG · 2024-08-10
> > **Thanks for your reply.**
> >
> > I am happy with the authors' reply and will keep my score.

---

> > > ### Author Response · Authors · 2024-08-12
> > > **Official Comment by Authors**
> > >
> > > Thanks for your positive comment Reviewer rEuG. We are happy to answer your questions and concerns.

---

### Author Rebuttal · Authors · 2024-08-05

Hello Reviewers, we appreciate you taking the time to read and evaluate our paper. It is great to see that you liked our paper and found it impactful.

We would like to summarise the updates we made to the paper and extra experiments we have done to address your concerns:

1. **Changes to the paper**: We included all the comments in the main text and updated the manuscript with a blue color in our draft. These updates primarily concern clarifying experiment setups and fixing typos.

2. **New results with new data type**: We have added a set of experiments with the MXFP4 format which is a similar format to FP4. Our results show that applying the randomized Hadamard transformations makes the input distribution more uniform than Gaussian, which is suitable for integer quantization (see our reply to the reviewer hgDj)

3. **New results with Mixtral**: we ran QuaRot on Mixtral 8x7B v0.1, a mixture of expert model, to show that QuaRot also applies to this architecture. We ran weight-only (W4A16) and 4-bit weights and 4- and 6-bit activations and performed within 1.5% of the baseline model on the LM eval benchmarks using QuaRot, highlighting the method’s validity on this different architecture.

| Format          | PPL  | Avg acc |
|-----------------|------|---------|
| FP16 (baseline) | 3.84 | 77.63%  |
| W4A16KV16 128G  | 3.95 | 77.51%  |
| W4A6KV16 16G    | 3.92 | 77.41%  |
| W4A4KV16 16G    | 4.18 | 76.33%  |


4. **Answering other questions**: We also addressed all questions about the paper in different sections.

Thanks again for your time. We look forward to discussing and answering follow-up questions and comments.

---

### Decision · Program_Chairs · 2024-09-25

**Decision:**

Accept (poster)

**Comment:**

This paper presents QuaRot, a framework for quantization of large language models (LLMs) using Hadamard transformations applied to weights and activations. QuaRot is able to quantize the entire model, including activations and caches, in 4 bits without significant loss in performance. The approach builds on concepts from existing works, such as incoherence processing and computational invariance, to achieve significant improvements in computational efficiency and memory usage while maintaining strong performance in language generation and zero-shot tasks. Reviewers acknowledged the method's effectiveness, particularly in improving quantization without major performance loss. However, there were concerns about the novelty of the contribution, as the paper primarily integrates methods from previous research. Additionally, some aspects, such as the motivation for using Hadamard matrices and the comparison to other quantization techniques, were found to be insufficiently detailed. Most of them have been addressed during the rebuttal period. Overall, the practical benefits and solid experimental results make this paper a valuable contribution to the field of LLM quantization. I recommend accepting the paper.